# Search for Structural Basis of Interactions of Biogenic Amines with Human TAAR1 and TAAR6 Receptors

**DOI:** 10.3390/ijms23010209

**Published:** 2021-12-25

**Authors:** Anna V. Glyakina, Constantine D. Pavlov, Julia V. Sopova, Raul R. Gainetdinov, Elena I. Leonova, Oxana V. Galzitskaya

**Affiliations:** 1Institute of Mathematical Problems of Biology, Russian Academy of Sciences, Keldysh Institute of Applied Mathematics, Russian Academy of Sciences, 142290 Pushchino, Moscow Region, Russia; quark777a@gmail.com; 2Institute of Protein Research, Russian Academy of Sciences, 142290 Pushchino, Moscow Region, Russia; constpavliii@gmail.com; 3Institute of Translational Biomedicine, St. Petersburg State University, 199034 St. Petersburg, Russia; y.sopova@spbu.ru (J.V.S.); gainetdinov.raul@gmail.com (R.R.G.); 4Laboratory of Amyloid Biology, St. Petersburg State University, 199034 St. Petersburg, Russia; 5Animal Genetic Technologies Department, University of Science and Technology, 1 Olympic Ave, 354340 Sochi, Russia; 6Institute of Theoretical and Experimental Biophysics, Russian Academy of Sciences, 142290 Pushchino, Moscow Region, Russia

**Keywords:** β-phenylethylamine, tyramine, gamma-aminobutyric acid (GABA), cadaverine, putrescine, trace amines, trace amine receptors, TAAR6, TAAR1

## Abstract

The identification and characterization of ligand-receptor binding sites are important for drug development. Trace amine-associated receptors (TAARs, members of the class A GPCR family) can interact with different biogenic amines and their metabolites, but the structural basis for their recognition by the TAARs is not well understood. In this work, we have revealed for the first time a group of conserved motifs (fingerprints) characterizing TAARs and studied the docking of aromatic (β-phenylethylamine, tyramine) and aliphatic (putrescine and cadaverine) ligands, including gamma-aminobutyric acid, with human TAAR1 and TAAR6 receptors. We have identified orthosteric binding sites for TAAR1 (Asp68, Asp102, Asp284) and TAAR6 (Asp78, Asp112, Asp202). By analyzing the binding results of 7500 structures, we determined that putrescine and cadaverine bind to TAAR1 at one site, Asp68 + Asp102, and to TAAR6 at two sites, Asp78 + Asp112 and Asp112 + Asp202. Tyramine binds to TAAR6 at the same two sites as putrescine and cadaverine and does not bind to TAAR1 at the selected Asp residues. β-Phenylethylamine and gamma-aminobutyric acid do not bind to the TAAR1 and TAAR6 receptors at the selected Asp residues. The search for ligands targeting allosteric and orthosteric sites of TAARs has excellent pharmaceutical potential.

## 1. Introduction

Biogenic amines belong to the group of organic substances with high biological activity. They are formed in living organisms by decarboxylation of amino acids by decarboxylase enzymes. Biogenic amines include dopamine, norepinephrine, adrenaline, serotonin, melatonin, tryptamine, histamine, and others. In animals, many biogenic amines act as hormones and neurotransmitters. Biogenic amines play essential roles in growth, proliferation, differentiation, cell migration, regulation of genes, synthesis of proteins and nucleic acids, and regulation of ion channels [1]. Polyamines are an integral part of cellular and genetic metabolism and aid in transcription, translation, signaling, and post-translational modifications [2].

Trace amines are endogenous compounds present in mammalian tissues at very low (nanomolar) concentrations. According to their classification [3], trace amines include amines of a predominantly aromatic nature, the concentration of which is less than 100 nM. Trace amines have long been ignored in human pathology due to the difficulty of their detection in biological fluids. In the human body, trace amines play an important role in synaptic signaling in the central nervous system, i.e., act as neurotransmitters or neuromodulators [4]. Currently, the role of TAAR-associated biogenic amines as neurotransmitters or neuromodulators in vertebrates during synaptic transmission in the central nervous system is well studied [5,6,7,8]. However, TAARs have also been identified in other types of cells, including cells of the immune system [5]. For example, the binding of classical trace amines of phenylethylamine and tyramine to TAAR1 stimulated the migration of granulocytes, induced the secretion of cytokines by T cells, and of immunoglobulins by B cells [9]. There is evidence that the impaired metabolism of biogenic amines may be associated with autoimmune disorders and cancer. In many forms of cancer, cadaverine and putrescine concentrations increase in cancer cells and blood plasma, urine, and feces [10,11,12,13,14,15,16,17]. In patients with leukemia at the initial stage, the levels of cadaverine and putrescine in the urine were markedly increased. During the period of remission, they returned to normal values, and in patients with actively recurrent leukemia, on the contrary, increased even more [12]. Prior to the discovery of TAAR, the receptors for cadaverine and putrescine had not been identified. Recent studies have shown that altering the gut microbiome can influence carcinogenesis in the gut and organs distant from the gut via microbiome-derived cadaverine, which acts on TAAR1, TAAR8, and TAAR9 [18]. Increased concentrations of biogenic amines in the blood and urine of cancer patients may reflect the increased activity of enzymes responsible for their synthesis. In addition to the de novo synthesis of biogenic amines, they can enter the body through fermented food or plants and be synthesized by the intestinal microbiome using bacterial decarboxylases. It is known that *Escherichia coli* bacteria possess a mechanism of resistance to the low pH of the external environment due to decarboxylation of lysine (product is cadaverine) and ornithine (product is putrescine). An increase in the level of cadaverine, for example, in ulcerative colitis, can occur in response to an increase in acidity in the colon or changes in the species composition of the bacterial population in the intestine. It should be noted that only ornithine decarboxylases are present in mammalian cells, while lysine decarboxylases are absent [19]. Cellular ornithine decarboxylase decarboxylates ornithine to form putrescine. There is an assumption that ornithine decarboxylase is responsible for the decarboxylation of lysine in mammals [20], but it has been shown that the inhibition of ornithine decarboxylase does not affect the synthesis of cadaverine [21]. To date, the question of the presence of an enzyme responsible for the synthesis of cadaverine in mammalian cells remains open. Numerous studies have shown that inhibition of ornithine decarboxylase decreases the rate of tumor growth and increases the antitumor activity of T cells [22]. An increased concentration of putrescine increases the ability of cancer cells to metastasize, decreasing the antitumor functions of immune cells [23]. An increase in the level of cadaverine in the body of a cancer patient, on the contrary, promotes a mild course of the disease. In patients with early breast cancer, a decrease in the level of *E. coli* lysine decarboxylase was found. An increased level of lysine decarboxylase has been correlated with an increase in survival among patients with early breast cancer [18]. It can be assumed that cadaverine is toxic to cancer cells. However, toxicity has been shown to require very high concentrations, much higher than those found in cancer. Cell culture studies have shown that cadaverine at concentrations present in the body is non-toxic [18,24]. Thus, structurally similar amines have opposite effects on cancer development in mammals.

Although biogenic amines are important signaling molecules, the structural basis for their recognition by TAAR is not well understood. TAARs belong to the class A G protein-coupled receptor family (A-GPCR), including rhodopsin [5]. A-GPCR is characterized by two types of ligand binding sites: allosteric and orthosteric. The first consists mainly of unstructured extracellular regions of the protein molecule, and the second is formed by transmembrane helices [25]. Ligands that bind to allosteric sites act as modulators and can include compounds with a wide range of activity, such as positive allosteric modulators (PAMs). PAMs increase the receptor response to the orthosteric ligand; negative allosteric modulators (NAM), which reduce receptor sensitivity, and neutral allosteric ligands, which bind to the allosteric site but do not affect the interaction with the orthosteric ligand [26]. However, allosteric modulators of TAARs are not currently described.

In zebrafish *Danio rerio*, TAAR13c has been shown to be responsible for the specific, highly sensitive recognition of cadaverine [27]. Further studies revealed an orthosteric binding site for TAAR13c and cadaverine (Asp^3.32^ (TAAR13c Asp112) + Asp^5.42^ (TAAR13c Asp202)) [28]. In humans, six functional TAARs (TAAR1, TAAR2, TAAR5, TAAR6, TAAR8, and TAAR9) are known to exist, all of which demonstrate significant structural similarity to TAAR13c [29,30]. In 2018, computer simulations predicted the same orthosteric binding site for Asp^3.32^ + Asp^5.42^ with cadaverine and putrescine for TAAR6 and TAAR8 [31]. Interestingly, cadaverine can induce the uptake of Ca^2+^ by the cell line HIBCPP of human vascular plexus cancer [30]. A detailed study has shown that the ability of cadaverine to induce the Ca^2+^ response depends on TAAR1 expression in the cells of the choroid plexus of the human ventricle [27,30]. Based on these data taken together, it can be expected that other TAARs can be activated by cadaverine and putrescine.

In the development of new therapeutic agents, a special role is played by studying the spatial interaction of the ligand and the receptor. However, TAARs are present in most cells in small amounts at low levels; therefore, data from experimental structures are not currently available in the database.

In this work, we analyzed the amino acid sequences of trace amine-associated receptors in vertebrates, compared the tertiary structures of TAAR1 and TAAR6 predicted using the GPCR-I-TASSER program [32], and characterized the allosteric pockets of TAARs and potential orthosteric binding sites for ligands: β-phenylethylamine, tyramine, cadaverine, putrescine, and gamma-aminobutyric acid (GABA). Understanding the mechanism of TAAR binding to ligands targeting allosteric and orthosteric sites has great pharmaceutical potential.

## 2. Results

### 2.1. Bioinformatics Analysis of Amino Acid Sequences of TAAR Proteins in Different Animal Species

Bioinformatics analysis of amino acid sequences of TAAR proteins in different animal species showed that TAARs represent a well-defined coherent protein family closely related to the GPCR family. If we consider the TAAR family as independent, then the amino acid sequence of these proteins should be distinguished by their unique motifs (fingerprints), which determine the belonging to this family of receptors. Usually, these motifs do not overlap in the primary structure and are spaced apart in the sequence, but they can be brought together in the tertiary structure. Using the PRINTS Database program (http://130.88.97.239/cgi-bin/dbbrowser/fingerPRINTScan/FPScan_fam.cgi (accessed on 26 August 2021)), we identified a group of conservative motives (fingerprints) characteristic of TAAR (Figure 1, Table 1). The PRINTS program for isolating families is based on the principle of searching for motifs in all databases, taking into account the discrepancy between individual amino acid residues, both within one motif and the whole motifs in the entire sequence. Thus, when an unknown protein is identified, the probability of detecting its paralogs increases.

The spatial structures of the TAAR1 and TAAR6 receptors obtained using the GPCR-I-TASSER program [32] clearly show that the motifs are located both on the outer surface of the membrane (fingerprints 2 and 3) and on the inner side of the membrane surface, i.e., inside the cell (fingerprint 1) (Figure 2). These data suggest that motifs play an important role in ligand binding and signaling into the cell.

It is known that GPCRs from different subfamilies can bind orthosteric ligands through aspartic acid residues in transmembrane α-helices III (canonical Asp^3.32^) or V (noncanonical Asp^5.42^), while diamine receptors contain both aspartates. Most mammalian TAARs contain Asp^3.32^, and any mutation in this position disrupts the recognition of trace amines [28]. Using multiple sequence alignment, we found the TAAR-conservative positions of Asp^2.50^, Asp^3.32^, Asp^3.49^, Asp^5.42^ and Asp^6.59^. In mice and rats, in TAAR7a, Asp^3.32^ was replaced by Glu^3.32^ (Figure 1).

Additionally, using multiple alignments, we showed the presence of canonical aspartic and glutamic acid residues located in the extracellular domains of TAAR1 and TAAR6 (Figure 2). According to our hypothesis, these amino acid residues form a site for allosteric interaction with ligands on TAARs. It is known that, in A-GPCRs, the sites of allosteric binding to ligands are located on the membrane surface in the unstructured region of the protein [33]. In TAARs, this pocket is supplemented with aspartic and glutamic acid residues, which are included in the group of conserved motifs (fingerprints) located on the membrane’s outer surface (Figure 1 and Figure 2).

### 2.2. Amino Acid Sequence Identity of Human, Mouse, and Fish TAARs

Different mammals have different numbers of TAARs. To date, the largest number (26) has been found in the flying fox (*Pteropus*), and not a single TAAR has been found in the bottlenose dolphin (*Tursiops truncatus*) [34]. Humans have nine TAARs, only six of which (TAAR1, TAAR2, TAAR5, TAAR6, TAAR8, and TAAR9) are functional [34,35,36,37,38,39]. Using the BLAST program, it was determined to what extent the human TAARs are identical to each other (see Table 2). 

From the data in Table 2, it can be concluded that there are two groups of the most similar TAARs. The first is TAAR1 and TAAR2 (more than 50% identity), and the second is TAAR6, TAAR8, and TAAR9 (more than 67% identity). This is in part consistent with the phylogenetic analysis that divides TAAR into three groups. The first group includes TAAR1-TAAR4. These TAARs are considered the oldest, most conservative; they developed under negative evolutionary pressure and, in most families, are represented mainly by one isoform [34,40,41,42]. The second group includes TAAR5-TAAR9, which have changed under positive evolutionary pressure and are represented in families by several isoforms [34,41,42]. Finally, the third group includes TAARs, which are specific for teleosts. Recently, they have undergone evolutionary spread [40], which may be caused by a change in the ligand binding site [28].

From the data in Table 3, it can be concluded that the identity between the human and mouse TAAR1 receptors is high (75%). The same is observed for TAAR6 receptors (88%). At the same time, the identity of the human and mouse TAAR1 and TAAR6 receptors with the fish TAAR13c receptor does not exceed 45% (see Table 3).

### 2.3. Docking of Ligands with Human TAAR1 and TAAR6 Receptors

The structures of human TAAR1 and TAAR6 receptors were predicted using the GPCR-I-TASSER program [32]. Five models were obtained for each receptor. In this work, we examined five ligands: β-phenylethylamine, tyramine, putrescine, cadaverine, and GABA.

After analyzing the amino acid sequences of the receptor structures, we selected three amino acid residues of Asp68(^2.50^), Asp102(^3.32^), Asp284(^6.61^) in the TAAR1 receptor and Asp78(^2.50^), Asp112(^3.32^), Asp202(^5.42^) in the TAAR6 receptor, which can be associated with the active center (see Figure 3).

Then the triplets BP^TAAR1^_1_{Asp68(^2.50^), Asp102(^3.32^), Asp284(^6.61^)} and BP^TAAR6^_1_{Asp78(^2.50^), Asp112(^3.32^), Asp202(^5.42^)} or pairs BP^TAAR1^_2_{Asp68(^2.50^), Asp102(^3.32^)}, BP^TAAR1^_3_{Asp102(^3.32^), Asp284(^6.61^)}, BP^TAAR1^_4_{Asp68(^2.50^), Asp284(^6.61^)} and BP^TAAR6^_2_{Asp78(^2.50^), Asp112(^3.32^)}, BP^TAAR6^_3_{Asp112(^3.32^), Asp202(^5.42^)}, BP^TAAR6^_4_{Asp78(^2.50^),Asp202(^5.42^)} of these amino acid residues were used as input data for the Galaxy7TM program [44]. From one computational experiment for each model for TAAR1 or TAAR6 + ligand + binding pocket (triplets or pairs of amino acids), 10 structures were obtained. For each model for TAAR1 or TAAR6 + ligand + binding pocket (triplets or pairs of amino acids), three independent computational experiments were carried out. Thus, after considering five models, five ligands, and five calculation options for the binding pockets, 7500 structures for TAAR1 and TAAR6 were obtained. The binding of the ligand to the receptor was determined by the presence of hydrogen bonds between them. A series of computational experiments has shown that no ligand forms hydrogen bonds simultaneously with the three preferred aspartic acid residues (see Figure 4A,B).

No ligands showed simultaneous hydrogen bonds with residues Asp68(^2.50^) and Asp284(^6.61^) for TAAR1 and residues Asp78(^2.50^) and Asp202(^5.42^) for TAAR6 receptors (see Figure 4). At the same time, all ligands were characterized by the formation of at least one hydrogen bond with Asp102(^3.32^) for TAAR1 and Asp112(^3.32^) for TAAR6 receptors (see Figure 5).

For all TAAR1 models, cases of simultaneous formation of hydrogen bonds between putrescine and Asp68(^2.50^) and Asp102(^3.32^), as well as for cadaverine with the same residues, were found. This observation was also true for Asp78(^2.50^) and Asp112(^3.32^) for TAAR6. The formation of hydrogen bonds between the pair of residues Asp112(^3.32^) and Asp202(^5.42^) and tyramine, putrescine, and cadaverine was observed for TAAR6 that was not characteristic for TAAR1.

Thus, it was obtained that for the aliphatic ligands (cadaverine and putrescine) and the TAAR1 receptor, there is one Asp68(^2.50^) + Asp102(^3.32^) binding site, and the TAAR6 receptor has two binding sites: Asp78(^2.50^) + Asp112(^3.32^) and Asp112(^3.32^) + Asp202(^5.42^). The same two binding sites of the TAAR6 receptor are observed for the aromatic ligand tyramine. For β-phenylethylamine and GABA, there is no binding to the selected residues of the TAAR1 and TAAR6 receptors (see Figure 4 and Table 4).

The following results were obtained when ligands docked with TAAR1 and TAAR6 receptors without determining the amino acid residues included in the binding pocket. When the TAAR1 receptor is docked with putrescine and cadaverine, one Asp68(^2.50^) + Asp102(^3.32^) binding site is observed and only for three and two of the five models, respectively (see Figure 6A). Furthermore, for TAAR6 with tyramine, putrescine, and cadaverine, one binding site Asp78(^2.50^) + Asp112(^3.32^) is observed and only for one of the five models considered for the receptor (see Figure 6B).

Thus, by analyzing the molecular docking between five ligands and the five obtained models, we assumed that the presence of two orthosteric sites characterizes the interaction with aliphatic ligands for both receptors. The first site for TAAR1 and TAAR6 is common and formed by two Asp^2.50^/^3.32^, while the second sites are different. The second site for TAAR1 is a combination of Asp^3.32^ and Asp^6.61^. The second site for TAAR6 Asp^3.32^ and Asp^5.42^ coincided with the site of interaction of cadaverine with the *Danio rerio* TAAR13c protein and was previously predicted for cadaverine binding to human TAAR6 [27,31]. The analysis of molecular docking data showed that cadaverine and putrescine could bind only to one orthosteric site Asp^2.50^/^3.32^ of the TAAR1 receptor, while TAAR6 can bind to two sites, Asp^2.50^/^3.32^ and Asp^3.32^/^5.42^ (Figure 7).

It is interesting to note that, in addition to the aspartic acid residues considered here, other amino acid residues also participated in the interaction with the ligands. The frequency of hydrogen bonding with such amino acid residues sometimes even exceeded the frequency with Asp residues. The frequencies of formation (in%) of hydrogen bonds by various amino acid residues based on 3750 computational docking structures for each receptor are presented in Table 5. In the TAAR1 receptor, Ser106 is on the same helix as Asp102; Asn293 and Ser294 are on the same helix as Asp284. In the TAAR6 receptor, Ser205 is located on the same helix as Asp202. 

In all considered cases of interactions of ligands with receptors, the lowest frequency of hydrogen bonding is characteristic of β-phenylethylamine (between β-phenylethylamine and the considered Asp residues of the receptor, no two hydrogen bonds are formed simultaneously) and slightly higher for tyramine (see Figure 4). It is known that mainly hydrophilic amino acid residues are involved in forming hydrogen bonds, while this pair of ligands has an aromatic ring and is hydrophobic. Thus, it is assumed that docking of these ligands is less thermodynamically favorable than the docking of aliphatic ligands. The better binding of tyramine to the receptor in comparison with β-phenylethylamine is justified by the presence of a charged hydroxyl group at the opposite end of the molecule, which, apparently, increases the frequency of formation of hydrogen bonds with the receptor due to the appearance of an atom with high electronegativity in the system. Further studies are necessary to define the molecular mechanisms of the activation of TAAR1 by β-phenylethylamine.

## 3. Discussion

Most TAARs (TAAR2–TAAR9), including TAAR6, are traditionally considered as olfactory receptors responsible for detecting socially significant instinctive odors, but accumulating evidence suggests their function beyond the olfactory system as well [5,7,8]. The typical “death smell”, characteristic of the volatile biogenic amines cadaverine and putrescine, is an essential signal for survival. Cadaverine and putrescine belong to the group of biogenic amines found from bacteria to eukaryotes. However, unlike the widespread biogenic amines such as histamine, serotonin, dopamine, and adrenaline, the molecular basis and physiological effects of cadaverine and putrescine are still largely unknown. Receptors that bind to cadaverine and putrescine were discovered only recently in 2013 [27]. Our research has shown that cadaverine and putrescine can potentially bind to any TAAR, but with different affinities. The binding affinity will depend on the organization of receptors on the cell membrane, and, given the ability of TAARs to dimerize [5], TAARs are likely to form not only homo- and heteromers but also high-order oligomers in lipid rafts. Thus, depending on the composition of the oligomers, allosteric pockets of TAARs can cause various conformational changes in the transmembrane domains, opening specific orthosteric sites. This fine-tuning can allow cells to respond quickly to external physiological changes.

According to numerous studies, various representatives of GPCRs can form dimers for binding to ligands [45,46]. Dimerization can occur due to the formation of disulfide bridges by cysteine residues located at the N-terminus of the protein’s amino acid sequence. By multiple sequence alignment of various human, mouse, and TAAR13c *Danio rerio* proteins, we identified two highly conserved cysteine residues located at the N-terminus of the protein molecule. Moreover, mouse TAAR1 has only one cysteine residue in this region. Therefore, we hypothesize that these cysteines are responsible for the formation of both homo and heterodimers, as well as higher-order oligomers (Figure 2).

In addition, absolutely all TAARs are characterized by the presence of cysteine residues at positions 97, 105, and 190, as shown in the example of TAAR6 (Figure 1). For other TAARs, these numbers may differ, but all of them will occupy an exclusively definite place in the spatial structure of the receptor. In this case, the formation of a disulfide bridge between Cys105 in the third helix with Cys190 located between the fourth and fifth helices is typical for many other representatives of GPCRs [47,48]. It is assumed that this disulfide bridge plays an important role in stabilizing the structure of GPCRs. Determination of the functional role of highly conserved Cys97 requires additional research.

## 4. Materials and Methods

### 4.1. Analysis of the Sequences of Genes of the TAAR Family in Animals

Amino acid sequences of TAAR proteins were generated from the UniProt database. Multiple sequence alignment was obtained using the program ClustalW (http://www.genome.jp/tools-bin/clustalw (accessed on 26 August 2021)). Further, the results obtained in the clutalw.aln format were imported into the GeneDoc program (Figure 1). The group of conservative motives (fingerprints) characteristic of TAARs was determined using the PRINTS Database program (http://130.88.97.239/cgi-bin/dbbrowser/fingerPRINTScan/FPScan_fam.cgi (accessed on 26 August 2021)).

### 4.2. Calculation of Amino Acid Sequence Identities for Human, Mouse, and Fish TAARs 

The amino acid sequence identities of the human, mouse, and fish TAARs were calculated using the BLAST program (https://blast.ncbi.nlm.nih.gov/Blast.cgi?PAGE=Proteins&PROGRAM=blastp&BLAST_PROGRAMS=blastp&PAGE_TYPE=BlastSearch&BLAST_SPEC=blast2seq&DATABASE=n/a&QUERY=&SUBJECTS= (accessed on 26 August 2021)) to align the two amino acid sequences using standard parameters. Amino acid sequences for human, mouse, and fish TAARs were taken from the UniProtKB database (https://www.uniprot.org (accessed on 26 August 2021)).

### 4.3. Modeling the Interaction of Human TAAR1 and TAAR6 Receptors with Ligands

Since there are no allowed spatial structures for the human TAAR1 and TAAR6 receptors, these structures were predicted using the program GPCR-I-TASSER (https://zhanglab.ccmb.med.umich.edu/GPCR-I-TASSER (accessed on 26 August 2021)) [32]. This program was specially developed for modeling the spatial structure of GPCR proteins based on the analysis of amino acid sequences of related proteins with an already known structure present in the Protein Data Bank (PDB). For each receptor, five models obtained with the GPCR-I-TASSER program were considered. Five models were obtained for each receptor. The Cα RMSDs for these structures are shown in Table 6.

For the TAAR1 receptor, in this parameter (Cα RMSD), models three and four differ most of all, and for the TAAR6 receptor, models two and five. It should be noted that model five differs from other models by the absence of disulfide bond between Cys95 and Cys181 for TAAR1 and Cys105 and Cys190 for TAAR6.

To simulate potential interactions of human TAAR1 and TAAR6 receptors with ligands (β-phenylethylamine, tyramine, putrescine, cadaverine, and gamma-aminobutyric acid, see Table 7), the Galaxy7TM service, which was developed by the Computational Biology Laboratory of Seoul University as part of the GalaxyWEB server [44], was used.

For each receptor, five models were considered with which ligands with preferential binding at aspartic acid residues interacted, based on experimental data [28]. The calculation options for each model are listed in Table 8. As a control, docking of receptors with ligands was carried out without determining the amino acid residues included in the binding pocket. Each of the computational experiments was performed three times to improve the accuracy of the obtained results.

The amino acid residue numbers in the sequences of different TAARs may not be the same, but they will be located in a certain place on the helix. Therefore, we used the Ballesteros–Weinstein amino acid residue numbering system, proposed specifically for the GPCR, where the first digit denotes the helix number and the second digit denotes the amino acid residue number [49].

Thus, for each of the five models of human TAAR1 and TAAR6 receptors, docking was carried out with five ligands (β-phenylethylamine, tyramine, putrescine, cadaverine, and gamma-aminobutyric acid) and triplets or pairs of aspartic amino acid residues were taken as a binding pocket. For each model receptor + ligand + binding pocket (pair or triplet of aspartic acids), 30 structures were obtained. As a result, 3750 structures in total were obtained for each receptor.

The binding of the ligand to the receptor was determined by the presence of hydrogen bonds between the aspartic acid residues of the receptor included in the binding pocket and the ligand.

## 5. Conclusions

In this work, we found a group of conserved motifs (fingerprints) that form the allosteric pocket of TAARs. Computational experiments have shown that cadaverine, putrescine, and tyramine can bind at two sites (Asp78(^2.50^) +Asp112(^3.32^) and Asp112(^3.32^) + Asp202(^5.42^)) with the human TAAR6 receptor. Moreover, cadaverine and putrescine can bind only in one site (Asp68(^2.50^) + Asp102(^3.32^)) with the human TAAR1 receptor, while β-phenylethylamine and GABA do not bind at all to these receptors at the selected Asp residues. The two orthosteric sites that we have described for cadaverine and putrescine will become accessible in a space-dependent manner. Thus, we can assume that cadaverine and putrescine can bind to any member of the TAAR family. However, the specific effects of TAAR1 and TAAR6 activation are still not well understood, and many important issues require further study.

## Figures and Tables

**Figure 1 ijms-23-00209-f001:**
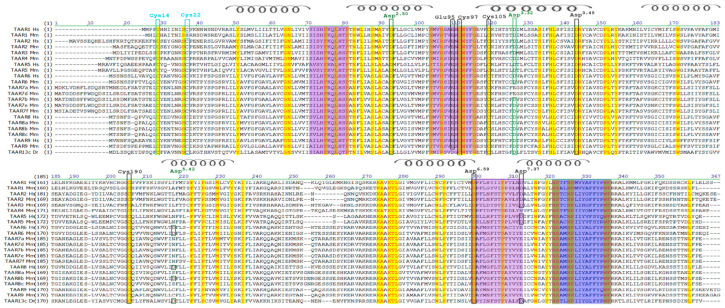
Comparison of amino acid sequences of mouse, human, and fish TAAR proteins. The group of conservative motifs (fingerprints) is highlighted in purple, and the motive that determines belonging to the rhodopsin family, predicted using the PRINTS Database program, is highlighted in blue. Conserved residues of aspartic (Asp^2.50^, Asp^3.32^, Asp^3.49^, Asp^5.42^, Asp^6.59^, Asp^7.37^) and glutamic (Glu95) acids were noted. In addition, the numbering of cysteine residues is given for the TAAR6 protein.

**Figure 2 ijms-23-00209-f002:**
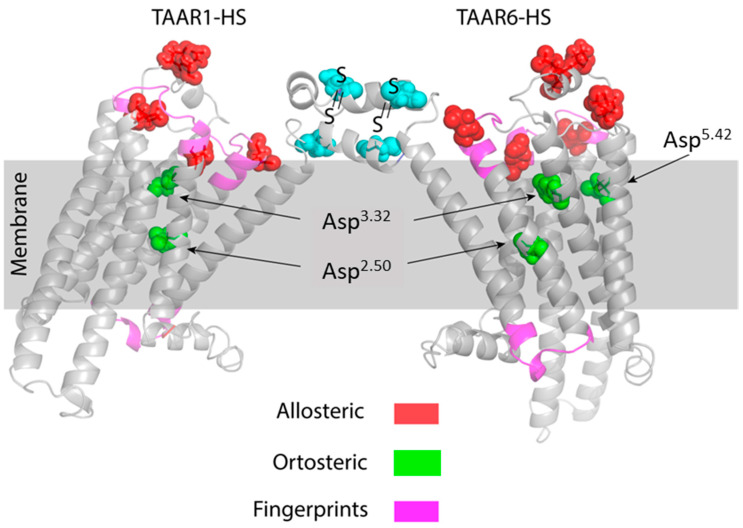
Spatial structures of the human TAAR1 and TAAR6 receptors were predicted using the GPCR-I-TASSER program [32]. The group of conservative motives (fingerprints) predicted using the PRINTS Database program is highlighted in magenta. Potential allosteric binding sites for positively charged ligands TAAR1 and TAAR6 on the membrane surface are highlighted in red; orthosteric binding sites for aliphatic ligands are shown in green. Possible S–S disulfide bridges in the presumed TAAR1 and TAAR6 heterodimer are highlighted in cyan. All crystal structures were visualized with PyMol.

**Figure 3 ijms-23-00209-f003:**
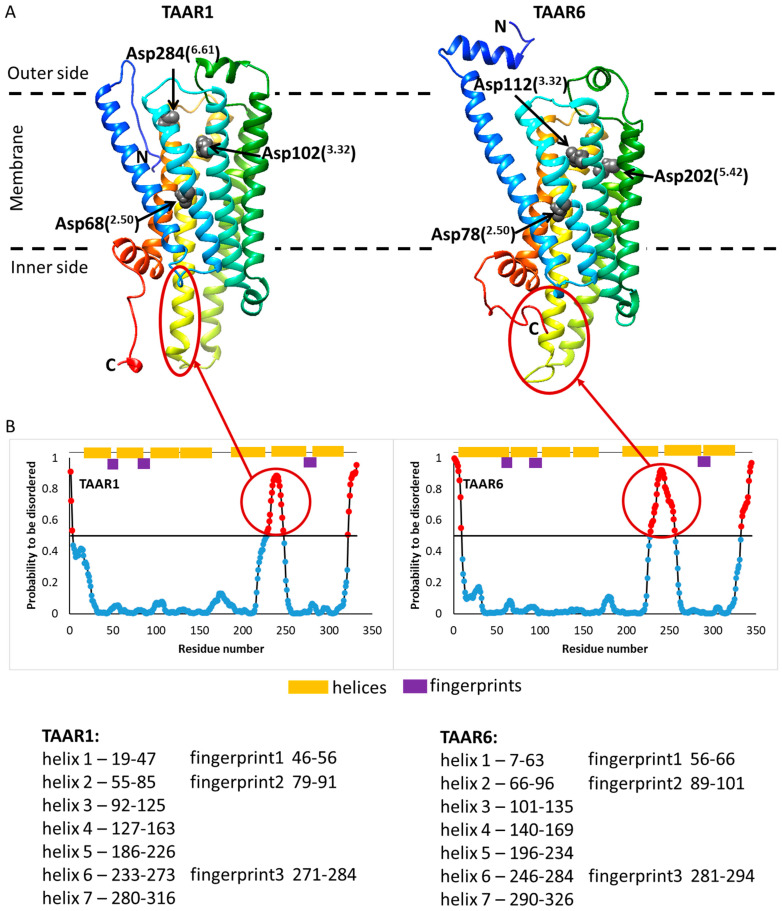
Spatial structures of the TAAR1 and TAAR6 receptors. Amino acid residues Asp68(^2.50^), Asp102(^3.32^), Asp284(^6.61^) for TAAR1 and Asp78(^2.50^), Asp112(^3.32^), Asp202(^5.42^) for TAAR6, which are probably included in the active center of these receptors, are highlighted in gray (**A**). Disordered regions were calculated using the IsUnstruct program [43] (**B**). Numeration of helices is given according to model 1 for both TAARs.

**Figure 4 ijms-23-00209-f004:**
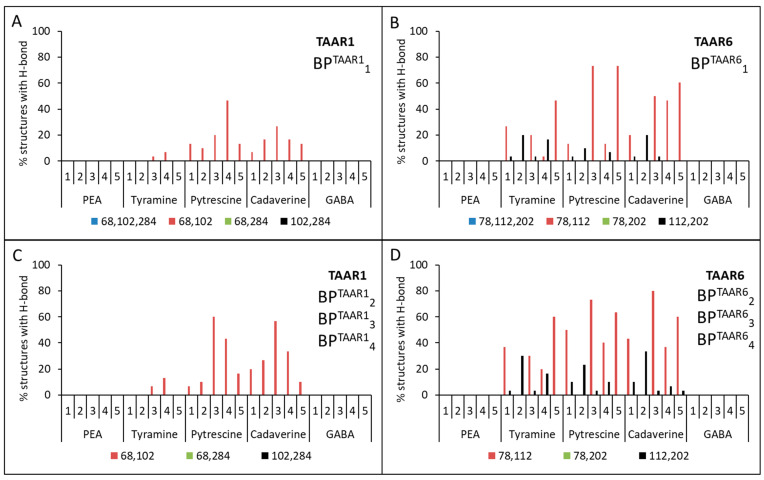
Percentage of structures in which the formation of two hydrogen bonds is observed simultaneously between ligands and the corresponding aspartic amino acids of the receptors. A series of computational experiments when triplets BP^TAAR1^_1_ and BP^TAAR6^_1_ (**A**,**B**) and pairs of aspartic amino acids, BP^TAAR1^_2_, BP^TAAR1^_3_, BP^TAAR1^_4_ (**C**) and BP^TAAR6^_2_, BP^TAAR6^_3_, BP^TAAR6^_4_ (**D**), were included in the binding pocket.

**Figure 5 ijms-23-00209-f005:**
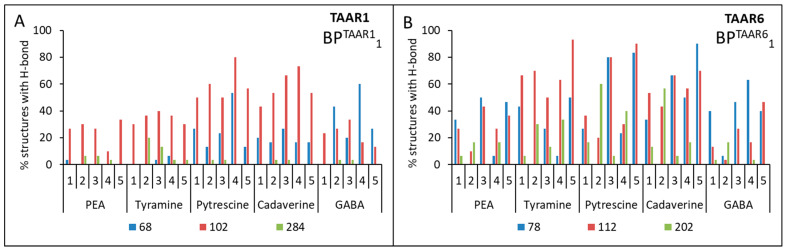
Percentage of structures in which the formation of hydrogen bonds between ligands and the corresponding aspartic amino acids of the receptors is observed. A series of computational experiments when triplets of aspartic amino acids BP^TAAR1^_1_ (**A**) and BP^TAAR6^_1_ (**B**) were included in the binding pocket.

**Figure 6 ijms-23-00209-f006:**
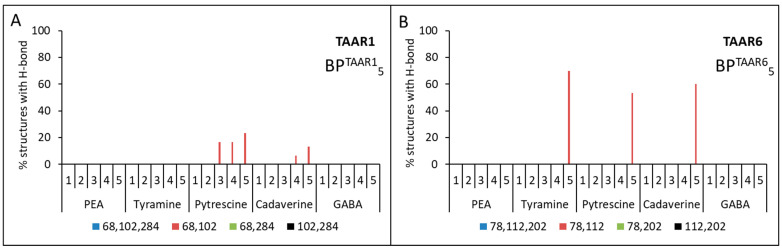
Percentage of structures in which the formation of two hydrogen bonds is observed simultaneously between ligands and the corresponding Asp residues of the receptors in a series of computational experiments when no binding pocket BP^TAAR1^_5_ (**A**) and BP^TAAR6^_5_ (**B**) is specified.

**Figure 7 ijms-23-00209-f007:**
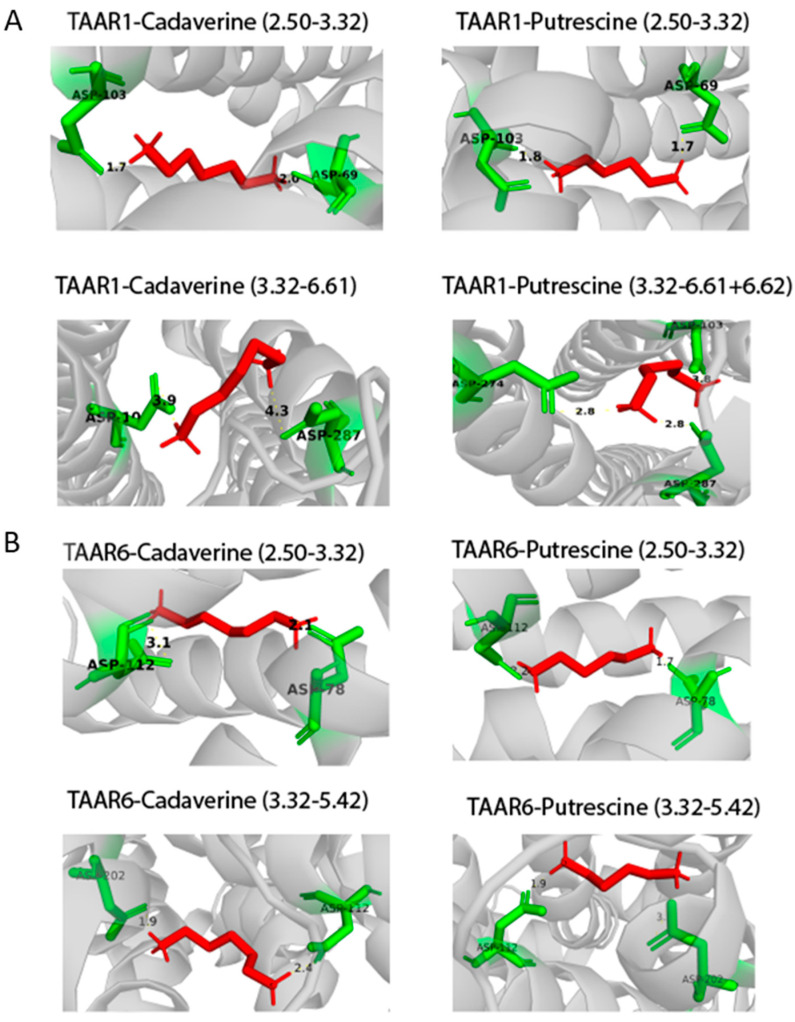
(**A**) Putative orthosteric binding sites of TAAR1 to aliphatic ligands. Cadaverine and putrescine can only bind to one orthosteric site of the Asp^2.50^/^3.32^ TAAR1 receptor. (**B**) Putative orthosteric binding sites of TAAR6 to aliphatic ligands. Cadaverine and putrescine can bind to two orthosteric sites, Asp^2.50^/^3.32^ and Asp^3.32^/^5.42^ of the TAAR6 receptor. All crystal structures were visualized with PyMol.

**Table 1 ijms-23-00209-t001:** Group of TAAR conservative motifs (fingerprint) of humans, mouse, and fish.

Name	TAAR Conservative Motifs		
	Fingerprint 1	Fingerprint 2	Fingerprint 3
TAAR1 *H. sapiens*	SISH**FK**Q**L**HT**P**** ***	**S**MVRSAEH**CW**YF**G**	**D**PFLHYII**P**PTLND
TAAR1 *M. musculus*	SISH**FK**Q**L**HT**P**	**S**MVRTVER**CW**YF**G**	**D**PFLGYVI**P**PSLND
TAAR2 *H. sapiens*	SISY**FK**Q**L**HT**P**	**S**MIRSVEN**CW**YF**G**	**D**PFLNFST**P**VVLFD
TAAR2 *M. musculus*	SISY**FK**Q**L**HT**P**	**S**MVRSVEN**CW**YF**G**	**D**PFLNFST**P**AVLFD
TAAR5 *H. sapiens*	AVSY**FK**A**L**HT**P**	**S**TIRSVES**CW**FF**G**	**D**SLLHFIT**P**PLVFD
TAAR5 *M. musculus*	AVSY**FK**V**L**HT**P**	**S**TVRSVES**CW**FF**G**	**D**SLLNFIT**P**PLVFD
TAAR6 *H. sapiens*	SI L H ** FK ** Q ** L ** HS ** P **	**S**MVRTVES**CW**YF**G**	**D**AFMGFIT**P**ACIYE
TAAR6 *M. musculus*	SIL H ** FK ** Q ** L ** HS ** P **	**S**MVRSIES**CW**YF**G**	**D**AFMGFIT**P**AYIYE
TAAR8 *H. sapiens*	SVLH**FK**Q**L**HS**P**	**S**MVRTVES**CW**YF**G**	**D**AFMGFLT**P**AYIYE
TAAR8 *M. musculus*	SVLH**FK**Q**L**HS**P**	**S**MVRSIES**CW**YF**G**	**D**AFMGFIT**P**AYVYE
TAAR8b *M. musculus*	SVLH**FK**Q**L**HS**P**	**S**MVRSIES**CW**YF**G**	**D**AFVGFIT**P**AYVYE
TAAR8c *M. musculus*	SVLH**FK**Q**L**HS**P**	**S**MVRSIES**CW**YF**G**	**D**AFMGFIT**P**AYVYE
TAAR9 *H. sapiens*	AILH**FK**Q**L**HT**P**	**S**TVRSVES**CW**YF**G**	**D**AYMNFIT**P**PYVYE
TAAR9 *M. musculus*	AILH**FK**Q**L**HT**P**	**S**TVRSVES**CW**YF**G**	**D**AYMNFIT**P**AYVYE
TAAR13c *D. rerio*	SIAH**FK**Q**L**QT**P**	**S**MIRSVDG**CW**YY**G**	**D**PYINFST**P**YALFD

* Amino acid residues that are identical in all studied proteins of the TAAR family are highlighted in bold; residues that are identical in the mouse and human TAAR6 proteins are underlined.

**Table 2 ijms-23-00209-t002:** Identity of human TAARs calculated using the BLAST program.

Human	TAAR1	TAAR2	TAAR5	TAAR6	TAAR8	TAAR9
TAAR1	100	51	39	41	39	43
TAAR2		100	43	39	37	40
TAAR5			100	44	44	45
TAAR6				100	80	69
TAAR8					100	68
TAAR9						100

**Table 3 ijms-23-00209-t003:** Identity of human and mouse TAAR1 and TAAR6 receptors calculated using the BLAST program.

	TAAR1 Mouse	TAAR6 Mouse	TAAR13c Fish
TAAR1 human	75	41	45
TAAR6 human	40	89	43
TAAR13c fish	44	42	

**Table 4 ijms-23-00209-t004:** Amino acid residues TAAR1 and TAAR6, which are included in the sites of interaction with ligands.

Ligand	TAAR1	TAAR6
cadaverine	1. Asp68(^2.50^) + Asp102(^3.32^)	1. Asp78(^2.50^) + Asp112(^3.32^) 2. Asp112(^3.32^) + Asp202(^5.42^)
putrescine	1. Asp68(^2.50^) + Asp102(^3.32^)	1. Asp78(^2.50^) + Asp112(^3.32^) 2. Asp112(^3.32^) + Asp202(^5.42^)
tyramine	-	1. Asp78(^2.50^) + Asp112(^3.32^) 2. Asp112(^3.32^) + Asp202(^5.42^)
β-phenylethylamine	-	-
GABA	-	-

**Table 5 ijms-23-00209-t005:** Frequencies of formation (in%) of hydrogen bonds by various amino acid residues for the TAAR1 and TAAR6 receptors.

TAAR1		TAAR6	
Asp68 (helix 2)	12	Asp78 (helix 2)	19
Asp102 (helix 3)	21	Asp112 (helix 3)	21
Asp284 (helix 7)	1	Asp202 (helix 5)	6
Ser106 (helix 3)	9	Ser205 (helix 5)	4
Trp261 (helix 6)	4	Trp271 (helix 6)	6
Asn293 (helix 7)	6	Asn303 (helix 7)	9
Ser294 (helix 7)	4	Ser304 (helix 7)	5

**Table 6 ijms-23-00209-t006:** Cα RMSD between TAAR1 (TAAR6) receptor models.

TAAR1 (TAAR6)	Model 1	Model 2	Model 3	Model 4	Model 5
Model 1	0	4.7 (4.4)	3 (1.7)	4.5 (2.5)	2 (4)
Model 2	0	0	4.8 (5)	2.1 (5.2)	5.3 (6.2)
Model 3	0	0	0	5.6 (3.7)	2.2 (3.8)
Model 4	0	0	0	0	5.5 (4.9)
Model 5	0	0	0	0	0

**Table 7 ijms-23-00209-t007:** Structures of ligands for docking with TAAR1 and TAAR6 receptors.

Ligand Type Name		Structure
aromatic	β-phenylethylamine	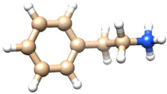
	tyramine	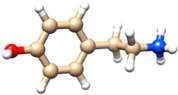
aliphatic	putrescine	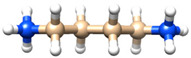
	cadaverine	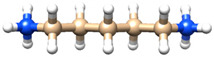
gamma-aminobutyric acid (GABA)		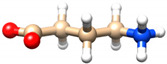

**Table 8 ijms-23-00209-t008:** Calculation options for the human TAAR1 and TAAR6 receptor models with Asp amino acid residues included in the binding pocket.

Calculation Option for Each Model	Aspartic Acid Residue Numbers	
	TAAR1	TAAR6
1	BP^TAAR1^_1_{Asp68(^2.50^), Asp102(^3.32^), Asp284 (^6.61^)}	BP^TA^^AR6^_1_{Asp78(^2.50^), Asp112(^3.32^), Asp202(^5.42^)}
2	BP^TAAR1^_2_{Asp68 (^2.50^), Asp102(^3.32^)}	BP^TAAR6^_2_{Asp78(^2.50^), Asp112(^3.32^)}
3	BP^TAAR1^_3_{Asp102(^3.32^), Asp284 (^6.61^)}	BP^TAAR6^_3_{Asp112(^3.32^), Asp202(^5.42^)}
4	BP^TAAR1^_4_{Asp68(^2.50^), Asp284 (^6.61^)}	BP^TAAR6^_4_{Asp78(^2^^.50^), Asp202(^5.42^)}
5	BP^TAAR1^_5_{–} ^1^	BP^TAAR6^_5_{–} ^1^

^1^ Computational experiments were performed without specifying preferred amino acid residues for the binding pocket.

## Data Availability

Not applicable.

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
