# Peer review of "Search for Structural Basis of Interactions of Biogenic Amines with Human TAAR1 and TAAR6 Receptors"

_ijms, 2021, doi:10.3390/ijms23010209_

Round 1
Reviewer 1 Report
The authors provide a revised version of their previous manuscript with major text changes. This definitely improves the manuscript and helps the reader to follow the author's thoughts.
From my point of view there are to week points that have not been adressed:
1) a robust validation of their dimer model.
2) any experimental validation of their results
I can only support this manuscript with limited enthusiasm due to the missing experimental backup, but don't want to hinder publication of this manuscript. If the editor decides that this manuscript fits the aims and scope of IJMS, it could be accepted.
Reviewer 2 Report
Thank you. The paper is well written.
I accept the manuscript in the present form.